# A Creative and Movement-Based Blended Intervention for Children in Outpatient Residential Care: A Mixed-Method, Multi-Center, Single-Arm Feasibility Trial

**DOI:** 10.3390/children10020207

**Published:** 2023-01-24

**Authors:** Susanne Birnkammer, Claudia Calvano

**Affiliations:** 1Clinical Child and Adolescent Psychology and Psychotherapy, Freie Universität Berlin, 14195 Berlin, Germany; 2Clinical Child and Adolescent Psychology, Christian-Albrechts-Universität zu Kiel, 24118 Kiel, Germany

**Keywords:** blended intervention, creative therapy, movement-oriented therapy, residential care, mental health app, children and adolescents, resources, resilience, COVID-19, pandemic

## Abstract

The COVID-19 pandemic led to psychological distress among children and adolescents. Due to multiple psychosocial burdens, the youth in residential care were especially exposed to an increased risk of mental health problems during the pandemic. In a multi-center, single-arm feasibility trial, *N* = 45 children and adolescents aged 7–14 years were allocated to a 6-week blended care intervention, conducted in six outpatient residential child welfare facilities. The intervention covered a once weekly face-to-face group session for guided creative (art therapy, drama therapy) and movement-oriented (children’s yoga, nature therapy) activities. This was accompanied by a resilience-oriented mental-health app. Feasibility and acceptance analyses covered app usage data and qualitative data. Effectiveness was determined by pre-post comparisons in quantitative data on psychological symptoms and resources. Further, subgroups for poorer treatment outcome were explored. The intervention and app were considered to be feasible and were accepted by residential staff and the children. No significant pre-post changes were found across quantitative outcomes. However, being female, being in current psychosocial crisis, a migration background, or a mentally ill parent were correlated with change in score of outcomes from baseline. These preliminary findings pave the way for future research on blended care interventions among at-risk children and adolescents.

## 1. Introduction

In 2019, the Coronavirus Disease (COVID-19) spread rapidly, which led to a global pandemic in 2020 that altered people’s habitual lives, social interactions, and psychological conditions. The pandemic-related mitigation measures initiated structural, social, and individual changes within the family and resulted in psychological distress for a significant number of parents and their children [1,2,3,4]. According to results of the representative CoPsy study among *n* = 1040 German children and adolescents, more than twice as many children and adolescents in Germany reported a worsening of their health-related quality of life during the COVID-19 pandemic compared to the data of children and adolescents in a pre-pandemic study [5]. According to a longitudinal study during the COVID-19 pandemic [6], low educational attainment, migrant backgrounds, parental mental illness, and crowded home spaces were identified as risk factors for an increased burden among children and adolescents. In line with these findings, particularly vulnerable groups are identified as those children and adolescents who experienced financial, household, and/or social disadvantages prior to the COVID-19 pandemic, such as children with migrant or refugee backgrounds, and children in residential care [7,8].

Children and adolescents who are placed in residential care groups were at increased risk of psychological, social, and behavioral problems during COVID-19 [7]. In a study among 10–17-year-old children and adolescents (*n* = 856) from residential groups in Spain, 41% felt lonely during the lockdown and reported low levels of subjective well-being, with more pronounced effects for girls [9]. Despite the knowledge on specific risk factors for mental distress during the COVID-19 pandemic, including female gender, migrant background, living in residential care settings, or parental mental health problems, interventions for high-risk populations during the COVID-19 pandemic are limited; up to now, the evaluation of one digital pilot program that focuses on increasing social-emotional learning among refugee children has been published [10]. To date, no intervention addresses psychological distress among children and adolescents during the COVID 19 pandemic by considering multiple risk factors.

The promotion of resilience in children and adolescents may reduce the risk of having psychological symptoms during the global pandemic and increase children’s quality of life [11]. Empirical studies conducted during the first COVID-19 phase from February to July 2020 identified various protective factors in children and adolescents for their psychological well-being. The expansion of a social network of family, peers, and school relationships [12], as well as certain attitudes such as optimism [13], have been identified as resilience promoting factors in cross-sectional studies conducted in China in February–April 2020. In support of these findings, a Portuguese trial among adolescents aged 12–18 years in residential care institutions indicated a positive relationship between the sense of coherence and emotional state during the first COVID-19 pandemic period [7]. An Israeli study identified social support and daily routines [14] as protective factors, and a study with 9–15-year-old US children indicated a positive effect of physical activity on mental health during the first COVID-19 phase [15]. This evidence underlines the potential of early resilience-oriented prevention programs for fostering child’s well-being and intrapersonal protective factors during the COVID-19-pandemic.

An especially appropriate form of enhancing intrapersonal protective factors are creative and movement-oriented interventions, given their symbolic and non-verbal characteristics [16]. In a systematic review of occupational and activity-based therapies for 5–18-year-old children and adolescents at risk of psychological distress, movement-based interventions like yoga and sports showed moderate-strong effects on mental health and social participation [17]. Facing the ongoing COVID-19 pandemic, the need for resilience-oriented prevention programs seems to be even more demanding. First studies provide support for the efficiency of creative and movement-oriented interventions for children and adolescents in pandemic conditions. During the Ebola pandemic in Liberia in 2015, a large community-based pre-post study among 870 children aged 3–18 years reported positive effects of a face-to-face intervention, with components of art therapy, play therapy, and yoga sessions, for reducing children’s psychological stress symptoms [16]. A recent online art therapy trial conducted during the COVID-19 pandemic in Canada reported effects in reducing inattention among 4th–5th grade students in an elementary school [18]. While these results are promising, the number of available studies is limited, and more research is needed.

Due to social distancing regulations, implementing a prevention program during a global pandemic might be challenging. Thus, the use of web- and mobile-based interventions was considered to be especially appropriate for the flexible fast-moving demands of the COVID-19 pandemic [19] and is a convenient opportunity to implement mental health interventions during a lockdown-period. The first pilot trials conducted during the COVID-19 pandemic support the feasibility and effectiveness of digital interventions for children and adolescents. Of note, the interventions vary highly with respect to content, specific objectives, outcomes, and design. Overall, results are promising; a 12-week Zoom-delivered neuroscience intervention aimed to strengthen awareness, resilience, and action through mindfulness and yoga was found to be accepted and to be feasible by 14–18-year-old American high school students [20]; however, key outcome measures did not significantly change between the intervention and control group. Another Zoom-delivered intervention to support social skills during the COVID-19 pandemic was shown to be effective in improving social affiliation, emotional well-being, and self-efficacy in 16–18-year-old adolescents from the UK, based on pre-post comparisons between the intervention and control group [21]. To reduce anxiety and digital eye discomfort in 12–14-year-old students during the homeschooling period in China, a live-streaming app intervention was implemented, which included an upload of the participant’s personal physical activity via video or photo and relaxation activities for the eyestrain [22]. The authors found a significant decrease in the anxiety score at the 2-week follow-up in the intervention group compared to the control group. The telehealth intervention iCOPE was designed to support 6–12-year-old children and adolescents in their COVID-19 related anxieties through psychoeducation, emotion regulation exercises, and developing coping strategies. In a pre-post comparison, it was feasible, accepted, and was shown to be effective in reducing social anxiety (Cohen’s *d* = 1.07) [23]. 

Taking the evidence on resilience-oriented digital interventions into account, three research gaps can be identified:

First, prior interventions for the support of resilience-related factors in children and adolescents during the COVID-19 pandemic primarily addressed adolescents older than 12 years, showing a gap for younger children. Second, previous interventions mainly included school-based or non-risk groups, leaving aside youths exposed to multiple risk factors. Third, while digital interventions showed very promising results, blended-care interventions with participant support through in-person assistance instead of a stand-alone digital intervention might be an especially promising approach to promote both participants’ adherence and the effectiveness of the intervention [24]. No blended care intervention has been implemented for children and adolescents during the COVID-19 pandemic so far. 

This study aimed to address these gaps in research. In this paper, we present results on the feasibility, acceptance, and effectiveness of a blended care intervention, covering both face-to-face sessions and an accompanying resilience-focused mental health app, with creative and movement-oriented approaches for children and young adolescents aged 7–14 years in residential outpatient care. Furthermore, special risk factors concerning the mental health of children and adolescents in outpatient residential facilities during the COVID-19 pandemic and their impact on the efficacy of the intervention were examined.

## 2. Materials and Methods

### 2.1. Setting

The blended intervention was conducted during 6 weeks within the participating facilities. The facilities were outpatient day care facilities in Berlin, which support psychosocially burdened families with educational assistance by accommodating the children. This comprises the support of social learning within the group, the supervision of school development, and parent-centered measures to ensure that the child can remain within the family. All facilities accommodated the children and adolescents on a day-care basis, meaning that the participants attended the facilities only during daytime periods. The facilities are non-profit and independent organizations in the field of child and youth welfare, funded by the government or by subsidies. The determination about the admission of a family or child to the respective facilities is made by the respective youth welfare office.

### 2.2. Design

The trial is a non-randomized, pre-post pilot trial for the evaluation of the feasibility, acceptance and effectiveness of a 6-week blended care intervention among at risk children aged 7–14 years in residential care settings. In the first and last week, the measurements T0 (pre intervention) and T1 (post intervention) were scheduled. 

A total of four instruments in the form of questionnaires and structured interviews were used for the children and adolescents. Assessment was conducted by a research assistant from the study team 1 week before (T0) and 1 week after (T1) the intervention within the facility and lasted approximately 60 min. The child questionnaires and interview are available in the Appendix A.

Residential staff also took part in the survey. The residential staff included pedagogical specialists such as social workers or educators, of whom one per facility participated in the study. The baseline questionnaires provided information on clinical aspects and the family history of the children and adolescents. Additionally, a structured interview was conducted on T1 to integrate the professional’s perspective on the blended care intervention in the facility. The questionnaires for residential staff can be found in the Appendix A

Beginning in week 1, the participants were able to use the app, providing them with a complementary thematic approach to the face-to-face activities for the remaining 5 weeks. During weeks 2–5, the therapeutic face-to-face activities proceeded in the facilities. 

The single-arm, multicenter, pre-post feasibility trial was conducted as part of a blended intervention project in outpatient residential settings for children and adolescents in Berlin, Germany. The project was funded by the German Children and Youth Foundation and the Federal Ministry for Family Affairs, Senior Citizens, Women and Youth (BMFSFJ). The trial has been approved by the Ethics Committee of Freie Universität Berlin, Germany (016/2022) and has been registered in the German clinical trial register (DRKS00029290). This trial is reported according to CONSORT guidelines [25].

The study flow chart can be found in Figure 1.

### 2.3. Participants

Children and adolescents between 7 and 14 years who returned the informed consent document signed by the parent(s) or legal guardian(s) and had basic reading and writing skills in German were included in the study. 

Exclusion criteria were the presence of acute mental stress (psychotic episode, suicidal tendency) and severe cognitive impairment. Possession of a smartphone was not necessary, as a tablet was allocated for the facility, on which the app could be accessed. This tablet could also be used if the children hold a smartphone model that has an older operating system than Android 10.0.

If the pedagogical specialist within the institutions did not participate in the study, this was not an exclusion criterion for the respective child. As the existence of or contact with parents was not assured for all children and adolescents, demographic and psychological information about participants and their family was requested by residential staff.

Recruited were *n* = 49 children and adolescents aged 7–14 years in 6 different residential care facilities in Berlin, Germany. The children and adolescents participated in groups of a maximum of eight participants for the face-to-face activities. Complete paired measurement time points were available from a total of *n* = 35 participants; additionally, data from *n* = 5 pedagogical specialists (educators, pedagogical management, social workers) as residential staff were collected at baseline and post measurements. 

### 2.4. Measures 

Given the mixed-methods design, multiple survey instruments, both quantitative and qualitative, were utilized. 

#### 2.4.1. Measures for Children and Adolescents

(1)Baseline sociodemographic variables and media usage. Besides demographic variables, child hobbies, and interests, media usage and changes in media consumption as a result of the pandemic were assessed at baseline T0. The items regarding media usage were: (1) Media/device usage (“Which of the following devices do you use most often? Rate the frequency from 1 (most often) to 4 (least often) for each of the following devices: cell phone—computer/laptop—tablet/game console”); (2) Daily smartphone usage (“On average, how much time do you spend on your smartphone per day? (<30 min—30–60 min—1–2 h—3–4 h—>5 h”)); (3) Change in media consumption since the pandemic (“How has your media consumption (= time spent on laptop, smartphone, or tablet) changed since the pandemic? (five-step response format: significantly more—significantly less”)); (4) Subjective feeling about own media consumption (“How do you feel about your amount of media consumption? (five-step response format: very good—very bad”)). The questionnaire is available in the Appendix A.(2)Child psychological symptoms. We used the Beck’s Youth Inventory (BYI-2) [26], a validated and manualized German version of the BYI-2 [27]. This questionnaire includes five scales, 20 items each: Depression (BDI-Y), Anxiety (BAI-Y), Anger (BANI-Y), Disruptive Behavior (BDBI-Y), and Self-worth (BSCI-Y). Participants rated the total of 100 items on a four-point Likert scale as to how often this applied to them in the last 2 weeks (“never—always”). The Cronbach Alpha of the various scales can be rated as good to very good for the baseline measurement (0.87–0.94) as well as for the post-measurement (0.86–0.94) within the current sample.(3)Intrapersonal and social resources. We used six subscales of the Resource Questionnaire for Children and Adolescents (FRKJ 8–16) [28], a validated and manualized German questionnaire, which is designed to assess the intrapersonal and social resources of children and adolescents. The six subscales capture intrapersonal resources (empathy and perspective-taking, self-efficacy, self-esteem, sense of coherence, optimism, and self-control) with a total of 36 items. Cronbach alpha reliabilities were adequate to good (0.67–0.80) in the baseline-survey as well as in the post-measurement (0.66–0.84) in the current sample.(4)Structured interview. The structured interview collects feedback on the app, on the face-to-face activities, and on the blended care intervention in general. For the app-related information, the children provided their current well-being after using the app, their liking of the design, as well as a possible recommendation to others. They also rate the activities (theater, nature, art, yoga) by favorability and give a preference between app or face-to-face activity. Finally, we asked the children if they would have preferred a longer project duration and what they like best about the project. The interview was only implemented in the T1 post measurement and can be found in the Appendix A.

#### 2.4.2. Instruments for Residential Staff 

(1)Child and parent-related risk factors. Using a self-generated questionnaire with 9 items, we obtained information regarding preexisting factors affecting the intervention execution, child-centered and parental risk factors regarding mental health distress during the COVID-19 pandemic, and participants’ residential care histories: (1) Child’s dyslexia (“Does the child or adolescent have a reading disability?”); (2) Child’s current crises or problems (“Are there currently problems/crises with the child/adolescent? (If yes, please state briefly)”); (3) Parental contact (“Does the child or adolescent have regular contact with parents?”); (4) Frequency of contact (“How often has the child/adolescent contact with at least one parent?”); as well as (5) History of connection to the facility (“How long does the child/adolescent attend the facility?”) and (6) History of connection to the youth welfare office (“How long has the child been involved with the youth welfare office?”) were obtained. Parent-centered information, such as (7) Migration background (“Do the parents have a migration background?”) and (8) Mental illness (“Does at least one parent or guardian of the child/adolescent have a mental illness?”) were also collected. Information to a respective DSM-5 diagnosis for the mentally ill person could be given if present (9). The questionnaire can be found in the Appendix A.(2)Structured Interview for the feasibility and acceptance of the blended care intervention. The 15 item-structured interview was conducted at T1 within the facility by a trained research assistant. Particularly, perceived strengths and suggestions for improvement of the intervention were explored, the app was evaluated by the staff, and feedback on the impression of children’s and adolescent´s acceptance of the intervention regarding usage and app design was collected. The interview of the residential staff is available in the Appendix A.

### 2.5. Blended Creative and Movement-Oriented Intervention

The “& Action!” project hybridized a 4-week face-to-face group intervention with a resilience-oriented mental health app, which conveyed psychological topics using both creative and movement-oriented therapy methods.

Using the overarching metaphor of a ship’s journey, the participants used drama therapy methods to learn about emotion regulation, nature education methods to learn about their self-worth, art therapy methods to learn about resource activation, and yoga therapy exercises to learn about recognizing their own limits. Each week, the participants practiced one therapy method both face-to-face and through exercises within the app. The face-to-face activities guided the participants and introduced the specific exercise, while the app is used to intensify and reflect the exercises in private. In addition, the app provided supplementary exercises on the topic, which can, however, be conducted independently. Table 1 shows an overview of the face-to-face and app content, the objectives, as well as the type of therapy.

#### 2.5.1. Procedure

In total, the intervention was conducted for 6 weeks, with data collection appointments in week 1 and 6 and a 4-week phase of 3-h face-to-face activities per week. In the first session of the intervention, the app and the implementing therapists were introduced, and the baseline measurement took place. Each of the six sessions was conducted on the same weekday at the same time of day. Participants had access to the app from the first introductory session and were able to conduct the digital exercises in line with the face-to-face exercises. The app modules were unlocked successively per week in order to enable simultaneous thematic treatment with the face-to-face activities. The app-based modules consumed approximately 5 min of actual smartphone usage daily. The remaining functions of the app, such as the diary, psychoeducation, and emergency exercises, were available at all times.

##### 2.5.2. Mondori App

The Mondori App is a resilience-oriented, gamified, web-based app with art-therapeutic, music-therapeutic, nature-therapeutic, theater-pedagogic, dance-therapeutic, and yoga-based cognitive-behavioral and positive psychological units for children and adolescents. The content of the Mondori app is based on recommendations and manuals for cognitive-behavioral therapies for children, specifically children within psychologically distressed family homes [29,30,31]. In addition, manual-based positive psychological content has been integrated into the modules, such as gratitude, mindfulness, and resource focus [32]. The app uses the metaphor of a ship’s journey and contains four modules plus one introductory module in the form of islands, which are to be explored by the children (see Figure 2). Each island addresses an overarching psychological topic using a playful method for each therapy type. The following contents are addressed per island/module: Setting goals and promoting motivation [Onboarding—Ship] (1), recognizing and accepting feelings [Jungle of emotions] (2), promoting self-esteem [Valley of self-love] (3), activating resources [Underwaterworld] (4), and recognizing and setting one’s own limits [Frosty landscape] (5). 

Integrated gamification elements encouraged participants to continue using the app; children and adolescents gained a seashell when completing a task. Furthermore, a regional flower seed on each island if all exercises on that island were completed is gained. This seed became planted in the ship’s yard and allowed to be nurtured by mindful questions, e.g., “What made you smile today?”, helping the plant flourish.

In addition to the islands, children can reflect about the activities or record thoughts and feelings in an embedded journal. Also, an SOS button in the shape of a life ring was embedded, which referred to emergency phone numbers for children and adolescents. Each island contains three challenges that the children accomplish by using a chat function to a sloth avatar. Prior to each challenge, a “requirement box” is used to indicate what the children need for the respective exercise. The program distinguishes between instant, daily, and weekly challenges, whereby only the instant challenge is performed exclusively on the smartphone. The daily and weekly challenges are guided and reflected on the smartphone, but the exercises are conducted in real life. In the daily challenges, mindful impulses are set, e.g., “Record sounds with your smartphone that calm you down today”. The weekly challenges sequentially rely on seven daily routines to build on each other. For example, each day children are encouraged to draw a picture with the emotion they felt most strongly that day. The challenges are approached using the congruent therapy method to face-to-face intervention.

In an initial pilot phase, the app was tested for functionality and usability with *n* = 10 children aged 8–13 years in a facility for children of mentally ill parents. The children were asked online about their experience with the app during use, and technical changes were made afterwards.

Participants of this trial utilized the free downloadable version of Mondori, which was available through the Apple Store and Google Play Store during the project period. The participants started in April 2022 with the app version (1.2.0). As of 15 May 2022, there had been automatic diary entries in a new version (1.3.0), in which a short summary of the exercise was automatically saved in the diary. Besides this, there were no other changes from version 1.2.2 to 1.3.0.

##### 2.5.3. Creative and Movement Based Face-to-Face Group Intervention

The four face-to-face units were conducted within the facilities and took place for 3 h per week for a total of 4 weeks. Reference to the app was given at the beginning and end of each activity: First, the previous app use was reflected and afterwards the motivation to use the app for the next week was reinforced.

The procedure of the activities followed a standardized manual, which was implemented in each institution. It was developed and conducted by the respective qualified and experienced therapists: A drama therapist, a nature pedagogue, an art therapist, and a children’s yoga teacher. 

The face-to-face activity content used the same metaphor as the app; the participants “traveled” with the therapists to a different island each week that they also encounter in the app. In addition, the type of therapy was maintained congruent within the app and face-to-face activity on a week-by-week basis.

Drama therapy was the initial face-to-face session and thematized the jungle of feelings. In the group, the connection between feelings and the body was linked through play and physical movement. 

Nature therapy followed as second session, exploring the “valley of self-love” with the aim of strengthening the participants’ self-esteem. The units took place in a greenspace in or close to each facility and commenced with a participatory group conversation about the concept of self-esteem. 

In the third session, art therapy was conducted with the participants in order to activate and strengthen the resources of the children and adolescents. The theme was “Underwater world”. 

The final session used yoga exercises to carry out the theme of boundaries. Thematically, the participants were now located at the “frost landscape” island. The perception and connection of the physical reactions of one’s own limits will be brought closer by means of breathing exercises, balance exercises (e.g., Om mantra, focus on exhalation (Rechaka)), in the form of asanas (e.g., crane (Bakasana), chair (Utkatasana), mountain hold (Tatsana)) and concentration exercises, all embedded in an ice world imaginary journey. 

### 2.6. Data Analysis

For data analysis, SPSS 29 was used. After screening for outliers and missing data (for further information, see Section 2.6.2), missing data were imputed, and the dependent variables were calculated by sum scores of the items from the respective questionnaires. Reliability analyses (Cronbach’s alpha as measure for internal consistency of the scales) were then computed among this sample for the BYI-2 and FRKJ 8–16 questionnaires. For the drop-out analyses, correlations with possible impact variables were determined, followed by a log-linear model of a Bayesian test, since the expected frequencies were < 5 at one value. Independent-samples *t*-tests were computed for between-group differences on specific risk variables for the baseline scores and for the change scores (pre-post difference score). In case of a normal distribution violation, the robust Mann-Whitney-U test was used. The Pearson correlation coefficient as the effect size was respectively calculated for the Mann-Whitney U test using the Z-test statistic. A dependent sample *t*-test was then calculated to assess the effectiveness of the intervention quantitatively. The evaluation of the feasibility and acceptance of the app, the face-to-face activities, as well as the blended intervention in general was performed through data from the questionnaires, technical server information on the app usage, as well as qualitative interviews with the participants and the professionals of each facility. Participants had the opportunity to evaluate the in-app instant, daily, and weekly challenges by clicking three different smileys after completing the challenges (See Figure 2). Interview data were transcribed using MAXQDATA 2022 and assigned based on participant codes. Congruent content topics were categorized by codes and assigned using a content text analysis. The most frequently mentioned categories are reported in the results. 

#### 2.6.1. Drop Out Analyses

In total, 22.2% (*n* = 10) of participants dropped out before the post-measurement. To examine a possible interaction of the drop-out in the post-measurement, the variables age, gender, dyslexia, mother language, type of institution, smartphone time and confidence, parental mental illness, and migration background as well as the outcome variables at baseline were considered. Contingency coefficients and point biserial correlations were calculated for the respective variables. After descriptive examination, Cramer’s statistic for the correlation for the presence of parental mental illness was 0.36 and significant (*p* = 0.018); therefore, a loglinear Bayesian test was executed to test the correlation between a drop out and the presence of parental mental illness. A significant relationship was found between the presence of a parental mental illness and a drop out (*X*² (1) = 5.60, *p* = 0.018). The Bayes factor supported the alternative hypothesis ($BF_{01}$ = 4.292). The odds ratio showed that the probability of a drop-out of the intervention of children with at least one mentally ill parent was 10 times higher than for children without mentally ill parents.

#### 2.6.2. Missing Data

Missing data contained single missing values within the respective questionnaires’ scales, complete questionnaires missing, and drop out to post measurement. The BYI-2 questionnaire was missing for *n* = 3 (6.6%) participants at baseline measurement and for *n* = 2 participants (4.4%) at post-measurement. Since the missing values for drop out at the post-measurement date are found to be associated with parental mental illness, MNAR is assumed and consequently the missing values for complete post-measurement questionnaires are not imputed. Hence, imputation was conducted solely for individual values of responded questionnaires.

Accordingly, the respective sample numbers of the baseline and post-measurements are reported in the results. Single missing items were imputed by Expectation Maximization (EM) with five iterations, after a prior screening for patterns regarding MAR.

## 3. Results

### 3.1. Baseline Data

#### 3.1.1. Sample Characteristics

The final sample included data from *N* = 45 participants with a mean age of 10.6 years (SD = 1.73), 57.8% were male (*n* = 26). Two (4.4%) of the participants reported a non-binary gender identity.

Information on the respective affiliations with the residential care facilities and sociodemographic, cultural, and psychological characteristics of the participants can be found in Table 2.

#### 3.1.2. Media Usage and Prior Experiences with Creative and Movement-based Activities

In the baseline measurement, participants were asked about their previous media use, which can be seen in Table 3. Children and adolescents were also asked about previous experience and preference regarding the activities (art, theater, yoga, and being in nature). In total, 80% (*n* = 36) had prior experience with at least one of the activities mentioned, and 33.3% (*n* = 15) had already tried at least three of the activities mentioned. The activities of painting and being in nature appealed most to the children at baseline. 

#### 3.1.3. Baseline Data on Child Psychological Symptoms and Resources

Baseline assessment data on internalizing and externalizing symptoms were mainly increased compared to a norm sample [27]. Mean scores on anxious symptoms were slightly elevated (*T* = 58.55), and mean scores on depression (*T* = 61.50), anger (*T* = 62.67), and disruptive behavior (*T* = 62.43) were moderately elevated, based on the test manual indications [27]. The sample’s mean scores on self-worth were below average (*T* = 43.09). Across the internal resource scales, the mean scores of the sample are in the lower average range, with stanine scores ranging from *ST* = 2.87 (self-efficacy and self-worth) to 2.96 (self-control).

#### 3.1.4. Risk Factors

Preidentified risk factors for mental health problems [5,6,7,8] were examined for group differences on baseline outcome variables. A significant gender difference was found in the mean scores on disruptive behavior and empathy at baseline, indicating that female participants, on average, had lower mean scores on disruptive behavior (exact Mann–Whitney U test: *U* = 111.00, *Z* = −2.16, *p* = 0.031, *r* = −0.35) and higher mean scores on empathy (*t*(41) = 2.28, *p* = 0.028, Cohen´s *d* = 0.712). Participants in a current crisis had significantly higher scores on the depression scale (*t*(37) = 2.07, *p* = 0.045, *d* = 0.67), disruptive behavior (exact Mann–Whitney U test: *U* = 101.500, *Z* = −2.51, *p* = 0.011, *r* = −0.40), and significantly lower scores in self-worth (*t*(40) = −2.82, *p* = 0.007, *d* = −0.88) than participants not in a current crisis. Participants with mentally ill parents had no significant difference in baseline scores compared to children without mentally ill parents. Participants with a migrant background had significantly lower mean scores on depression (exact Mann–Whitney U test: *U* = 96.500, *Z* = −2.24, *p* = 0.024, *r* = −0.37) and higher mean scores on resources empathy (*t*(38) = 2.80, *p* = 0.008, *d* = 0.89), self-efficacy (*t*(29) = 3.50, *p* = 0.002, *d* = 1.13), and coherence (*t*(38) = 2.29, *p* = 0.028, *d* = 0.72) at baseline than participants without a migrant background. Table 4 shows the mean scores and effect sizes of the risk group analyses for gender, current crisis, parental mental illness, and migration background.

### 3.2. Acceptance and Feasibility Results

#### Acceptance and Feasibility of the Mondori App

Usage times. The server data was supplied by the technical app developers. In particular, the dropout rate and time, the amount of time participants engaged with an app session, and the feedback within the app are reported. The average total utilization time of the participants was 33.21 min. An approximate overall utilization time of (3 challenges × 5 min × 5 islands) = 75 min was estimated for the completion of all challenges during the 6 weeks. Utilization times are equilibrated across participants in the study with *n* = 16 children and adolescents using the app for less than 10 min in total, including seven of whom only registered, *n* = 7 using the app between 10–19 min overall, *n* = 13 using the app between 20 and 59 min in total, and *n* = 11 used the app for more than 60 min (see Figure 3A).

Drop outs. The point at which participants dropped out of the app could be identified using server data provided about the last module they had started (see Figure 3B). *N* = 11 participants dropped out after the introduction module. Seventeen children and adolescents dropped out during the module of Jungle Island, three children and adolescents dropped out during the penultimate module, and one participant utilized the app until the final module.

Child ratings of the challenges within the app. A total of 14 ratings were submitted. Of these, 11 were happy smileys, two challenges were given a neutral smiley once (ship introduction, instant drama therapy challenge), and one was given a sad smiley (daily art therapy challenge). Nature therapy and children’s yoga as well as weekly challenges for all forms of therapy were not evaluated.

### 3.3. Qualitative Interviews Regarding Mondori App

Questions during the interview addressed the feelings of the participants after using the app, their liking of the app design, as well as specific content that the participants particularly liked or disliked. 

In total, 68% reported a positive feeling after using the Mondori app (“good/very good/better/okay/relaxed”), 15% reported negative feelings (“bad, strained”), 18% gave no response or claimed to have not used the app. For the app design, participants gave an average school grade of 2.3 (good) and described the app design in the interview as “age-appropriate” (*n* = 17), “good/cool/okay” (*n* = 8), and “in need of shortening the text” (*n* = 8). In response to questions regarding the three aspects of the app that participants liked best and least, the following categories were established based on responses, which can be seen in Table 5; 61% (*n* = 17) would like to continue using the app after the intervention and 55% (*n* = 16) of the participants would recommend the app to their acquaintances or friends.

The evaluation of the residential staff regarding the app was found comparable to the participants’ rating. The appealing design of the app (average rating was very good—good), the game-like character of the app through chatting and the islands, and its ability to track the children were rated positively. In common with the children and adolescents, the pedagogical specialists criticized the text length of the app together with the technical errors (e.g., “some buttons did not respond”). The residential staff considered the usage of the app within the facilities during the project as rather low, partly varying. A continuation of usage after the intervention was predicted by most of the residential staff for a few individual participants, which were mainly older. The app was generally rated as “sufficient” and 100% of the residential staff would recommend the Mondori App to others. 

#### Acceptance and Feasibility of the Face-to-Face Activities

Participation in activities fluctuated and decreased from week to week (see Figure 1), caused by absences due to COVID-19 illness (*n* = 6), varying days of care within facilities (*n* = 2), school commitments (*n* = 4) on these days, and/or unwillingness to participate (*n* = 1).

### 3.4. Qualitative Data Regarding Face-to-Face Activities

In the post-measurement interview, participants were asked what activity they liked best during the program, with the majority (61.4%, *n* = 21) citing art and/or yoga exclusively, 8.6% (*n* = 3) reported no activities, and 20.1% (*n* = 7) reported drama or nature separately or mentioned them along with yoga and art.

This is congruent with the statements of the residential staff during the interview at T1. Responding to the question of which activity the children enjoyed the most as perceived by the pedagogical specialist, yoga and art were mentioned most frequently: “Art, because it’s the easiest way for the children to be able to express themselves. And yoga, because of the physical movement.” (Pedagogical director of a participating facility).

The residential staff estimated drama most frequently as more challenging for the children to accomplish: “Drama, because the children have to go outside their comfort zone and they might be ashamed of this within the group” (primary educator of a participating facility).

In the feedback session of the activities, the participants also reflected and evaluated about the learning and the experiences of the activities. Quotes and general perceptions of acceptance for face-to-face activities can be found in Table 6.

#### Acceptance and Feasibility of a Blended Intervention

Participants and residential staff were asked about their perceptions of the blended intervention at the T1 interview. This included a potential preference for the app or the face-to-face activities as well as a request to extend the duration of the intervention. In total, 74.1% (*n* = 26) of the participants and 80% (*n* = 4) of the residential staff preferred face-to-face activities compared to the app. The residential staff indicated that the motivation of the children was enhanced due to the face-to-face session. In addition, 22.9% (*n* = 8) of the participants considered both parts of the blended intervention equally valuable. Furthermore, 57.1% (*n* = 18) of participants requested a longer duration of the intervention, and 8.6% (*n* = 3) were uncertain in this regard. The majority of the residential staff (80%, *n* = 4) would have liked the project to run longer.

Table 7 displays the categorized responses and quotes to the question regarding which aspects of the blended intervention the children, adolescents, and residential staff enjoyed the most.

### 3.5. Quantitative Effectiveness Outcomes

#### 3.5.1. Intervention Effectiveness Data on Child Psychological Symptoms and Resources

Dependent samples *t*-test showed no significant results for a difference in participants’ anxiety (*t*(30) = −0.32, *p* = 0.753), depression (*t*(30) = -0.03, *p* = 0.98), anger (*t*(30) = 1.89, *p* = 0.069), disruptive behaviors (*t*(30) = 1.33, *p* = 0.194), and self-esteem (*t*(30) = 1.95, *p* = 0.060). 

The inferential statistical results of the mean difference of the average levels of empathy (*t*(34) = 1.33, *p* = 0.192), self-efficacy (*t*(34) = −0.29, *p* = 0.77), self-worth (*t*(34) = 1.60, *p* = 0.12), sense of coherence (*t*(34) = 1.71, *p* = 0.10), optimism (*t*(34) = −0.09, *p* = 0.93), and self-control (*t*(34) = 2.00, *p* = 0.06) were not found to be significant. The hypotheses could not be supported. Table 8 shows the mean values and results of the *t*-test for paired samples.

#### 3.5.2. Effects of Risk Factors on Outcome Change Score

When examining the change scores between T0 and T1 with respect to the risk groups, significant differences were found for the variables gender, migration background, and parental mental illness. The female participants’ scores on depressive symptomatology decreased from T0 to T1, whereas the male participants’ scores increased (*t*(28) = 2.20, *p* = 0.036, *d*= 0.85). Among participants with mentally ill parents, scores on anger decreased significantly from T0 to T1 compared to participants without mentally ill parents (*t*(28) = −2.28, *p* = 0.031, *d* = −0.84). Among children with a migrant background, scores on resources (self-worth (*t*(28) = 3.00, *p* = 0.006, *d* = 1.10), empathy (*t*(31) = 2.54, *p* = 0.016, *d* = 0.89), optimism (exact Mann–Whitney U test: *U* = 58.500, *Z* = −2.08, *p* = 0.005, *r* = −0.49), self-efficacy (*t*(31) = 3.40, *p* = 0.002, *d* = 0.92), and sense of coherence (*t*(31) = 2.65, *p* = 0.012, *d* = 0.93) significantly worsened at T1 compared to children without a migrant background. Figure 4 displays the change scores from T0 to T1 within the subgroups gender (A), participants with and without parental mental illness (B), and migration background (C–G).

## 4. Discussion

To our knowledge, this is the first trial to implement and evaluate a blended intervention for children and young adolescents in outpatient residential care settings. This study aimed to examine an innovative combination of creative and movement-oriented therapy methods with a complementary mental health app for its feasibility and effectiveness towards child-centered, internalizing and externalizing symptoms and resources among children in residential care during the COVID-19 pandemic. We also aimed to characterize certain risk groups for higher psychological symptoms and poorer resources among this sample.

The qualitative results suggest that the Mondori app was perceived as helpful among the participants. The smartphone seemed to be an appropriate device for the participants, based on their usage habits assessed at baseline. Both the technical and qualitative data indicate that participants were engaged with the app. Nevertheless, the average time of app utilization during the 6-week intervention was recorded as only 30.21 min. Given a continuous usage and accomplishment of the app-internal challenges, a usage duration of approximately 75 min during the intervention period was expected. In this study, half of the participants reported using the smartphone for less than 2 h per day. Of note, many facilities exerted limits on the children’s media use in general. Therefore, it is possible that the usage of the Mondori app competed with other smartphone usage purposes during the limited time available. This assumption is underlined by research showing that during the COVID-19 pandemic, youths primarily used smartphones for communicative actions [33]. Based on the server data, it is apparent that many participants ceased using the app at an early stage within the program. It is conceivable that the app was tested first, but the motivation of continuous usage across the intervention was diminished. A decrease in app usage over time has also been found in other feasibility mobile-based intervention studies for children and adolescents [34,35]. However, in-app evaluation response was marginal, and interviews were more appropriate for acceptability assessment among this age group. Participants favored the avatar, certain features such as the diary, or gamification elements in the app. This is consistent with the appeal for interactive elements to foster user engagement and thus create a higher impact of digital interventions [36]. The age range from 7–14 years made it challenging to fit all needs of the sample; therefore, some of the participants mentioned that the challenges were too easy while some of them mentioned that the texts were too long. By shortening or auditively redesigning the texts, the usage time and adherence should potentially be increased. Although the participants’ media usage behavior was assessed, no information was gathered regarding their own smartphone possession or their experiences with and affinity for mental health apps. These previous experiences might add to the participants’ technologic and eHealth literacy. This, in turn, might beneficially impact the acceptance and usage of a mental health app [37,38]. Despite the low usage rate of the participants, the satisfaction rate and the feedback of the participants were mostly positive and comparable to the average scores of acceptability in a meta-analysis of mental health mobile apps for preadolescents and adolescents [38]. 

The face-to-face activities were found to be especially embraced by participants, most notably the art and yoga activities, which were familiar activities for the children and adolescents, that provided low-threshold access. The combination of externally motivating face-to-face activities and self-initiated app usage was well accepted by both participants and residential staff. This supports the appropriateness of a blended care approach within young adolescents and children.

Another aim of this paper was the analyses of intervention effectiveness with respect to psychological symptoms and resources. Quantitative analyses of pre-post measurement within the sample did not reveal significant differences regarding internalizing and externalizing symptoms and resources. This result is congruent with other findings on social skill training interventions for this target group [39,40]. It should be considered that children and adolescents in residential care facilities are more likely to have experienced mental health problems [41,42]. For a target group that was already highly psychosocially burdened, this intervention could have been not sufficiently intensive in terms of time and psychotherapeutic affiliation, and too unspecific with regard to certain problems. According to the data of the residential staff, half of the participants were experiencing a current crisis. Congruently, the sample in this study had elevated baseline scores regarding depression, anger, and disruptive behaviors, as well as decreased scores in self-worth than normative comparison samples derived from the manuals [27,28]. However, no screening for mental health problems was conducted on the children and adolescents, nor were they excluded if they were currently experiencing a crisis according to the residential staff. The crises stated by the pedagogical specialists ranged from current housing or school problems to acute episodes of mental illness and thus represent a large heterogeneous range in terms of severity and type of crisis. Therefore, providing an adequate intervention for this heterogeneous group of individuals with different needs and varying emotional distress is challenging [39]. Possible reasons for this result could also have origins within the study design. The sample size is too minor for small effects and should be supplemented by a control group for a clinically valid statement. Another potential explanation for the non-significant improvement of psychological symptoms could be the concurrent duration of the war in Ukraine during 2022. The omnipresent media images of the conflict in Europe, the large number of Ukrainian refugee families, and the participants own insecurity about the war process could have negatively affected their well-being during the intervention.

Another aim of this study was the description of the sample in terms of risk groups regarding higher psychological symptoms and poorer resources during the COVID-19 pandemic and the analyses of possible effects on the change of the outcome measures. While the pre-post comparison regarding the outcome variables within the sample was not significant, additional risk factors for psychological symptoms during the COVID-19 pandemic could be identified, which influenced the baseline of the outcome variables as well as on the change score of the outcome measure. First, associations between prior identified risk factors and the baseline scores regarding psychosocial distress such as gender [8], having mentally ill parents [6], migration background [7], and a current crisis, were examined. Except for current crisis, the hypothesized correlates could not be confirmed by the data. Being female, having a migration background, and parental mental illness could not be replicated as risk factors in this study based on the baseline assessment. At baseline measurement, female participants had lower scores on disruptive behavior and higher scores on empathy, which is in line with common literature [43,44]. Furthermore, concerning depressive symptomatology, female participants seemed to benefit more from the intervention than male participants. This finding is consistent with the results of a German study examining the effectiveness of a program to reduce depressive symptoms in 8th graders, where a positive intervention effect was only found among female adolescents [45]. Also, almost half of the participants in this sample had parents with mental illness, which might increase the likelihood of having own mental distress [46,47]. The effect of parental mental illness was demonstrated by the higher probability of dropping out at T1. The influence of family factors on children’s drop out rates has also been found in previous studies [48,49]. This finding implicates the importance of a systematic assessment of the family’ s health before starting the intervention and an increased sensitivity to a potential dropout rate of children with mentally ill parents. Further, the presence of a mentally ill parent went along with higher reduction in anger symptoms. This suggests that the resilience-oriented strategies delivered by the blended intervention were especially fruitful for those children with a mentally ill parent. We can only speculate about reasons for this effect: as the face-to-face program was provided in a standardized manner, the children might have used the app more often, implying a higher intervention dose. However, this assumption could not be tested as according to the data privacy restrictions, viewing individual app utilization data from the server in terms of certain variables or groups was not permitted. For a future study, it would therefore be advisable to specifically interview the children and adolescents about their app usage at home or in private. Supporting the presumption of the particular benefit of digital interventions within this subgroup, specific digital interventions are already being developed for adolescents with mentally ill parents [50,51]. The online intervention for young adults provides promising results for the participant’s depression, anxiety, and stress [51]; however, the role of anger has so far not been studied in other research. Having a migration background could not be replicated as a risk factor for psychological distress among this sample, consistent with studies conducted prior the COVID-19 pandemic [52,53]. Children with a migration background scored higher on internal resources already at baseline. One reason for this could be the development of internal resources as a consequence of a multitude of demanding life challenges; children with a migration background may experience those demanding life challenges more often than children without a migration background and thus have more opportunities to strengthen resilience [54]. While the improvement in internal resources scores only among children without a migration background could be explained by lower baseline scores, further research is needed to get an understanding of the decrease of resources among the subsample with migration background. 

However, it should be noted that these factors may be interrelated and are additionally influenced by socio-cultural factors. For instance, shame can be a socio-cultural factor that might influence reporting of psychological well-being, as shown in [55]. Thus, socio-cultural factors might have an additional influence on the study findings. As we did not assess further socio-cultural factors, the hypothesized relation could not be examined in this study.

It should also be noted that in Germany, by the summer of 2022, some regulations regarding the COVID-19 pandemic had already been repealed, no school closures were occurring at that time, and leisure programs were once again being offered, making comparison with first-phase studies of the pandemic challenging. Due to the small sample and the high drop out rate, the results might have a lower reliability and should be considered as preliminary findings. Future studies regarding blended intervention designs should include a larger sample, a randomized controlled design with a comparison group, and follow-up measurements to determine statistically valid conclusions regarding efficacy. This intervention included multiple therapeutical components and implementation formats. A next step would be to examine these components and delivery formats concerning their respective influence on the effectiveness using a dismantling study. Further research should also examine the role of socio-cultural aspects as well as technological and e-Mental health literacy on the feasibility and effectiveness of digital interventions for children and adolescents.

The blended care intervention was designed to address the increased need for flexible, available, and psychosocial support for at-risk children and adolescents during the COVID-19 pandemic. The present pilot study was able to contribute to the preventive residential care service of an at-risk group during COVID-19 pandemic. The findings further indicate that being female, having a current crisis, mentally ill parents, and migration background have an impact on the effectiveness and adherence of an intervention during the COVID-19 pandemic and should, therefore, be considered and addressed when planning a psychosocial intervention. Due to increased risk for drop out, children of mentally ill parents should be the focus and might need additional support for attrition.

Furthermore, the first findings on the feasibility and effectiveness of creative and movement-oriented therapies with a complementary mental health app could be shown. In addition, as current feasibility studies of digital interventions predominantly have an overreliance on adolescent samples rather than children [56], this pilot project makes a significant contribution to digital interventions for research on mental health apps targeting children under 14 years.

Nevertheless, children and adolescents are currently experiencing an accumulation of global crises such as climate change, the Ukraine war, and currently, the feminist revolution in Iran. A focus on youths coping in those global crises and the development and implementation of crisis-related interventions would be desirable in future research and clinical practice.

## 5. Patents

A trademark registration was filed with the German Patent and Trademark Office for the word mark “Mondori”, relating to lead class 41. 

## Figures and Tables

**Figure 1 children-10-00207-f001:**
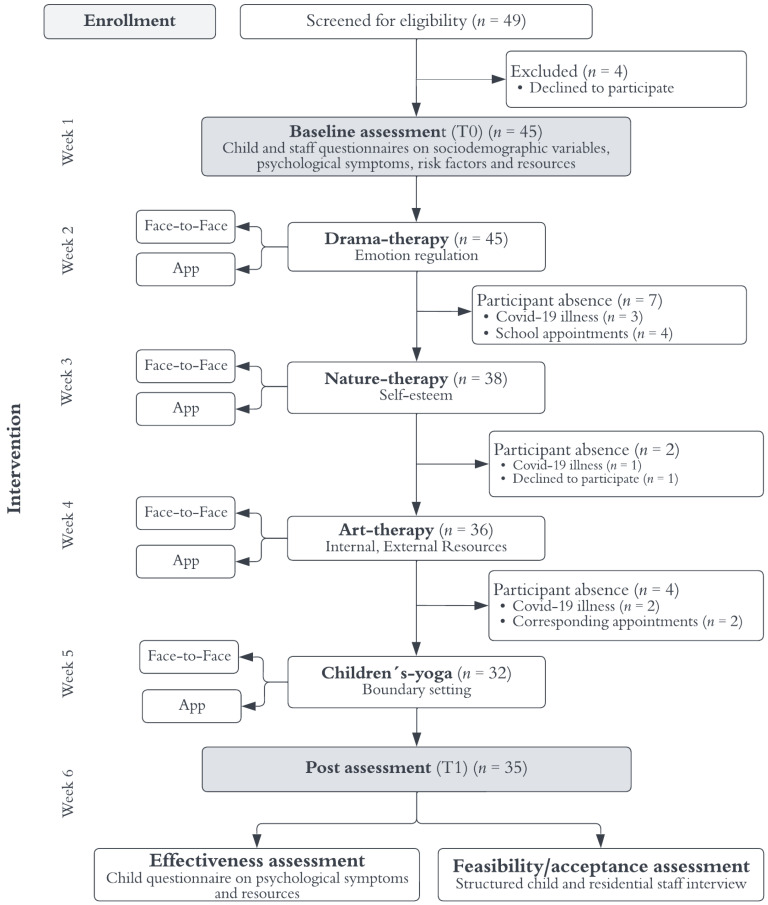
Flowchart of the study process with sample numbers within each activity and measurement data collection at T0 and T1. Participant absence describes the number of participants missing from the respective activity across all facilities.

**Figure 2 children-10-00207-f002:**
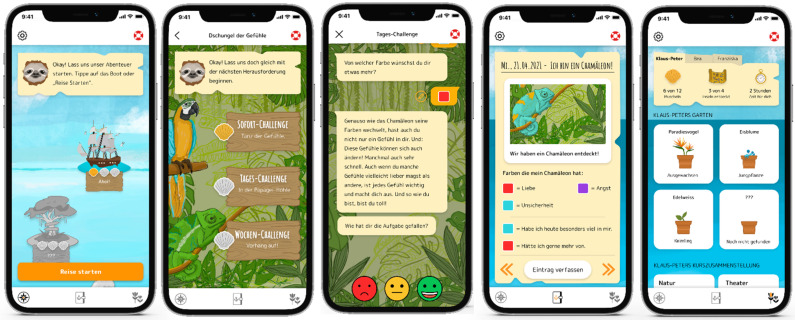
Screens of the Mondori App: Dashboard, challenge overview (jungle/week 1), challenge evaluation, diary, achievement statistics.

**Figure 3 children-10-00207-f003:**
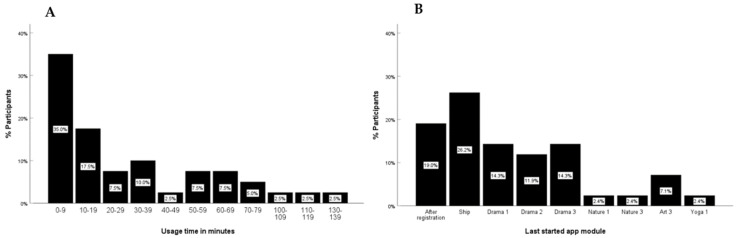
Server data for (**A**) app utilization per participant in minutes; 15% (*n* = 6) of the total sample used the app 0 min. (**B**) The last completed modules within the app per participant.

**Figure 4 children-10-00207-f004:**
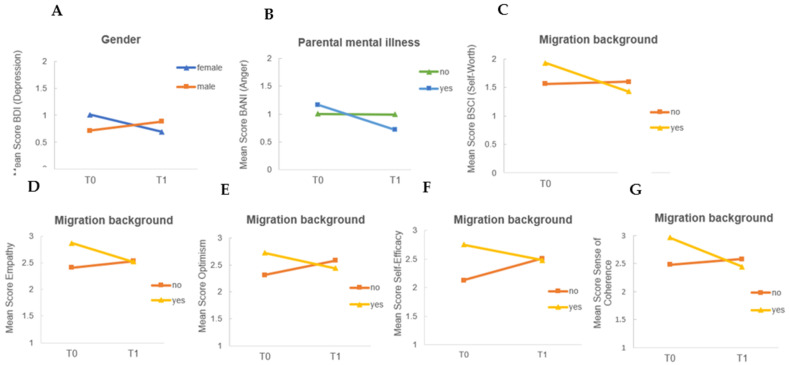
Change in outcome scores at T0 (pre intervention) and T1 (post intervention) regarding (**A**) depression scores in female and male participants, (**B**) anger scores in children with and participants without a parent with mental illness, (**C**) self-worth, (**D**) empathy, (**E**) optimism, (**F**) self-efficacy, and (**G**) sense of coherence in participants with and without a migration background.

**Table 1 children-10-00207-t001:** Blended care intervention per week regarding app and face-to-face content.

Week	Island	Objective	Therapy Type	App Content	Face-to-Face Content
1	Ship	Setting goals, motivation	-	Defining own goals, creating own motivation sentences	T0
2	Jungle	Emotion recognition and acceptance	Drama	Paraphrase emotions with body and face, writing a role play-script	Figure play with jungle animals, bodily and improvisational expression of emotions
3	Valley	Self-esteem	Nature	Encouraging braveness, meditation, seeing individual beauty of nature in oneself	Experience uniqueness of nature and create analogies with the self
4	Underwater	Activating resources	Art	Meditation, drawing internal and external resources	Meditation and creative creation of personal and external strengths
5	Ice	Setting boundaries	Yoga	Yoga-video, recognizing own boundaries, establishing rituals	Breathing exercises and asanas
6	-	Feedback and reflection	-	-	T1

**Table 2 children-10-00207-t002:** Sample characteristics (*n* = 45).

Variables	*n* (%)
Sociodemographic characteristics	
Mean age ± SD, years	10.6 ± 1.7
Male gender	26 (57.8)
Cultural characteristics ^1^	
German native language	26 (53.1)
Parental migration background	21 (42.9)
Psychological characteristics ^1^	
Presence of current crisis ^2^ (=yes)	25 (51.0)
Presence of reading disability (=yes)	14 (28.6)
Mentally ill parent(s) ^3^ (=yes)	21 (47.7)
Affiliation to residential care ^1^	
Duration of the previous residence	
in respective facility	
<1 year	14 (28.6)
1–2 years	13 (26.5)
2–6 years	14 (28.5)
>6 years	1 (2.0)
Other previous youth welfare facility	
<1 year	4 (8.2)
1–2 years	7 (14.3)
2–6 years	20 (40.8)
>6 years	9 (18.4)
Contact with parent(s) ^1^	
Yes	39 (79.6)
Frequency	
daily	18 (36.7)
weekly	6 (12.3)
monthly	10 (20.4)
annual	1 (2.0)

^1^ Data was obtained from information received from residential staff. ^2^ Crises were defined as current psychosocial problems that concern the child/adolescent. ^3^ This question should be answered based on the family record. The mental illness should have been diagnosed by a professional, a time frame of the diagnosis or information about therapy was not obtained.

**Table 3 children-10-00207-t003:** Media use and behavior of the sample.

Media Use and Media Consumption	*n* (%)
Most frequent device utilization	
Smartphone	31 (68.9)
Laptop	4 (8.9)
Tablet	5 (11.1)
Game console	2 (4.4)
Average time on smartphone/day	
<1 h	8 (22.2)
1–2 h	14 (38.9)
3–4 h	2 (5.6)
>5 h	12 (33.3)
Subjective app using confidence	
not confident (not, not at all confident)	6 (15.0)
partly confident	9 (22.5)
confident (confident, very confident)	25 (62.5)
App preferences (3 most mentioned)	
TikTok	14 (36.8)
YouTube	14 (36.8)
Roblox	12 (31.6)
Media consumption change ^1^	
a lot more	15 (37.5)
more	5 (12.5)
same	14 (35.0)
less	6 (15.0)
Consumption evaluation	
positive (good, very good)	27 (65.9))
neutral	10 (24.4)
poor (bad, very bad)	4 (9.7)

^1^ Since the beginning of the COVID-19 pandemic.

**Table 4 children-10-00207-t004:** Independent *t*-tests for potential risk factors gender, current crisis, parental mental illness, and migration background on the outcome measures at baseline.

	Gender				Current Crisis				Parental Mental Illness				Migration Background			
	Mean (SE)				Mean (SE)				Mean (SE)				Mean (SE)			
Outcome	**f** (*n* = 16)	**m** (*n* = 24)	*p*-value	d [95% CI]	**y** (*n* = 22)	***n*** (*n* = 17)	*p*-value	*d* [95% CI]	**y** (*n* = 20)	***n*** (*n* = 19)	*p*-value	*d* [95% CI]	**y** (*n* = 17)	***n*** (*n* = 20)	*p*-value	*d* [95% CI]
Self-worth	1.83 (0.15)	1.69 (0.11)	0.429	0.258 [−0.38–0.89]	1.70 (0.14)	1.78 (0.11)	0.689	−0.130 [0.76–0.50]	1.79 (0.13)	1.68 (0.14)	0.557	−0.190 [−0.82–0.44]	1.90 (0.14)	1.60 (0.12)	0.117	0.530 [−0.13–1.18]
Anxiety	1.08 (0.18)	0.875 (0.14)	0.357	0.301 [−0.34–0.93]	1.16 (0.17)	0.80 (0.14)	0.118	0.517 [−0.13–1.12]	1.13 (0.18)	0.87 (0.14)	0.252	−0.373 [−1.00–0.26]	0.85 (0.15)	1.12 (0.15)	0.253	−0.383 [−1.03–0.27]
Depression	1.02 (0.21)	0.72 (0.09)	0.207	0.472 [−0.17–1.11]	1.07 (0.16)	0.65 (0.11)	0.045 *	0.669 [0.01–1.31]	0.99 (0.43)	0.77 (0.10)	0.282	−0.35 [−0.98–0.29]	0.65 (0.14)	1.07 (0.15)	0.024 *	*r* = −0.37
Anger	1.08 (0.16)	1.06 (0.10)	0.940	0.024 [−0.61–0.66]	1.22 (0.13)	1.00 (0.13)	0.253	0.375 [−0.26–1.01]	1.27 (0.37)	0.98 (0.08)	0.117	−0.510 [−1.1–0.13]	0.95 (0.12)	1.25 (0.13)	0.097	−0.562 [−1.28–0.07]
Disruptive behavior	0.494 (0.11)	0.704 (0.07)	0.011 *	*r* = −0.40	0.74 (0.09)	0.48 (0.09)	0.031 *	*r* = −0.35	0.63 (0.10)	0.62 (0.09)	0.947	−0.021 [−0.65–0.61]	0.57 (0.09)	0.70 (0.10)	0.326	−0.329 [−0.98–0.32]
Empathy	2.91 (0.16)	2.45 (0.12)	0.028 *	0.712 [0.08–1.34]	2.61 (0.14)	2.60 (0.15)	0.965	0.014 [−0.60–0.62]	2.59 (0.65)	2.62 (0.14)	0.908	0.036 [−0.57–0.64]	2.89 (0.14)	2.35 (0.13)	0.008 *	0.886 [0.23–1.53]
Self-efficacy	2.57 (0.16)	2.48 (0.12)	0.658	0.139 [−0.47–0.75]	2.40 (0.14)	2.56 (0.14)	0.459	−0.233 [−0.84–0.38]	2.51 (0.13)	2.43 (0.15)	0.686	−0.126 [−0.73–0.48]	2.81 (0.16)	2.18 (0.09)	0.002 *	1.13 [0.46–1.80]
Self-worth	2.78 (0.16)	2.61 (0.14)	0.474	0.225 [−0.39–0.84]	2.37 (0.14)	2.97 (0.12)	0.007 *	−0.881 [−1.52–−0.23]	2.62 (0.15)	2.64 (0.17)	0.934	0.026 [−0.58 −0.63]	2.84 (0.17)	2.50 (0.15)	0.142	0.474 [−0.16–1.10]
Sense of coherence	2.76 (0.16)	2.64 (0.12)	0.537	0.194 [−0.42–0.81]	2.54 (0.12)	2.83 (0.13)	0.145	−0.463 [−1.08–0.16]	2.67 (0.13)	2.66 (0.15)	0.937	−0.025 [−0.63–0.58]	2.91 (0.15)	2.47 (0.13)	0.028 *	0.724 [0.08–1.36]
Optimism	2.68 (0.14)	2.41 (0.10)	0.123	0.491 [−0.13–1.1]	2.42 (0.12)	2.53 (0.13)	0.576	−0.176 [−0.79–0.44]	2.53 (0.13)	2.40 (0.12)	0.491	−0.215 [0.82–0.39]	2.68 (0.15)	2.33 (0.10)	0.052	0.635 [−0.01–1.27]
Self-control	2.82 (0.14)	2.60 (0.12)	0.254	0.361 [−0.26–0.97]	2.56 (0.13)	2.84 (0.13)	0.154	−0.453 [−1.07–0.17]	2.71 (0.12)	2.66 (0.16)	0.809	−0.075 [−0.68–0.53]	2.73 (0.17)	2.63 (0.11)	0.641	0.149 [−0.47–0.77]

Note: *d* = Cohen´s d, CI = Confidence interval of effect size d, f = female, m = male, y = yes, *n* = no. *r* = For the Mann-Whitney U test, the Pearson correlation coefficient *r* was calculated as an effect size measure. * significant for *p*-value = 0.05.

**Table 5 children-10-00207-t005:** Categorized responses to what participants liked best and worst in the app.

Supportive Feedback	Feedback Suggesting Improvement
Interaction with Avatar “chatting/interacting with Fauli”	Text length “too much texts/questions”
Aesthetics “graphics, design, islands, different worlds”	Presentation time “talked too fast”
App-features “rewards, diary, SOS-button”	Technical errors “pixelated animals”, “didn´t work”
Game character “different levels, content of challenges”	Content/Challenges “too easy, number of challenges”
Other	Presentation style
“nothing, app was not used”	“weird/not many answers”

**Table 6 children-10-00207-t006:** Qualitative assessment of face-to-face activities: General findings and quotes.

Therapy Type/Aim	General Findings	Quotes
Drama therapy/ Perception and regulation of emotions	Positive perception of expressing feelings nonverbally	“I discovered my body all new today—like a newborn! I am proud and happy” (10-year-old girl) “I can represent something with my body, too!” (11-year-old-boy) “I am happy right now but sad that the program is over.” (13-year-old non-binary participant) “I take with me today that role-playing is fun!” (9-year-old girl)
Nature therapy/ Enhancing self-esteem	By creating analogies to the animal world, participants were able to name their strengths in a simplified way:	“The bee is very diligent because it makes honey for people. And I’m very diligent because I did a lot of homework today.” (8-year-old girl) “A male and female unicorn met and fell in love with each other. So my strength is love.” (11-year-old boy) “That was pretty great, I don’t want it to end. And I really want to do this tree exercise again!” (9-year-old girl) “I like everything that has to do with nature.” (12-year-old boy)
Art therapy/ Activating resources	Participants especially seemed to enjoy the free and unassessed process of creating a painting and creating a strength animal or object to provide them with support:	“I am completely free!” (12-year-old girl) “This circle was particularly beautiful to me. I don’t know why, but when I look at it, I think of the future.” (13-year-old boy) “My magical creature, the unicorn, wants everyone to have happiness except the bad people.” (9-year-old girl) “I painted a very special panda.” (8-year-old girl) “These are lotus flowers with wings. They have a colorful power. If you put something in there, it becomes very small or very big.” (13-year-old girl)
Yoga therapy/Setting boundaries	Changing opinion during the session: At first, the participants were skeptical but then embraced the yoga session	“I want to remember everything I learned in yoga today and incorporate it into my life.” (11-year-old boy) “I’ve never seen the kids so happy. You really could hook them.” (Pedagogical specialist to yoga therapist)

**Table 7 children-10-00207-t007:** Most favored intervention components for children/adolescents and staff.

Intervention Part	General Findings	Quotes
Face-to-face activities	Movement (yoga) Freedom to create (art) Being outside (nature) Sympathy of the therapists	“The physical activity and releasing the energy in yoga was my highlight.” (11-year-old boy) “I liked that we were allowed to just paint away in art and didn’t have an objective.” (11-year-old boy) “I especially liked the fact that we were outdoors as well.” (10-year-old boy) “That the nature therapist was here.” (11-year-old girl) “The yoga teacher was very sympathetic to me.” (11-year-old boy)
Mondori app	Digital companionship of the avatar “Fauli”	“That Fauli was always with me.” (12-year-old boy)
Blended intervention	Voluntary character Multidimensional examination of topics	“That we could freely choose whether we wanted to attend.” (10-year-old girl) “The different approaches to the topic of emotions.” (Primary educator of a participating facility) “Creative and diversified change from everyday life.” (Pedagogical specialist of a facility) “Tasks were processed differently.” (Pedagogical director of a facility)

**Table 8 children-10-00207-t008:** Descriptive statistics and *t*-test for dependent samples.

Measure	Pre		Post		*p*-Value	95% *CI* for MD	*η*
	*n*	Mean (SE)	*n*	Mean (SE)			
Mental health							
Anxiety	42	0.99 (0.69)	34	0.97 (0.61)	0.776	[−0.24–0.19]	0.62
Depression	42	0.87 (0.64)	35	0.89 (0.63)	0.978	[−0.20–0.21]	0.64
Anger	42	1.10 (0.57)	34	0.93 (0.48)	0.071	[0.00–0.039]	0.56
Disruptive behavior	42	0.63 (0.42)	34	0.55 (0.32)	0.207	[−0.05–0.27]	0.46
Self-worth	42	1.71 (0.57)	35	1.49 (0.55)	0.060	[0.00−0.39]	0.54
Resources							
Empathy	45	2.60 (0.68)	35	2.54 (0.55)	0.191	[−0.04–0.30]	0.57
Self-efficacy	45	2.48 (0.64)	35	2.47 (0.48)	0.768	[−0.24–0.17]	0.66
Self-esteem	45	2.64 (0.72)	35	2.47 (0.74)	0.124	[−0.04–0.40]	0.70
Sense of coherence	45	2.66 (0.63)	35	2.50 (0.67)	0.077	[−0.03–0.45]	0.73
Optimism	45	2.48 (0.57)	35	2.51 (0.49)	0.931	[−0.21–0.19]	0.61
Self-control	45	2.66 (0.61)	35	2.39 (0.60)	0.061	[0.00–0.47]	0.71

Note: *η* = Cohen´s d, CI = Confidence interval of mean difference between pre and post mean.

## Data Availability

The data presented in this study are available on appropriate request from the corresponding author. The data are not publicly available as the privacy of the human subjects must be ensured.

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
