# Peer review of "A Creative and Movement-Based Blended Intervention for Children in Outpatient Residential Care: A Mixed-Method, Multi-Center, Single-Arm Feasibility Trial"

_children, 2023, doi:10.3390/children10020207_

Round 1

Reviewer 1 Report

Well designed paper.

However the sample size and high dropout rates make results less reliable.

Yoga interventions are not clearly mentioned. Being multicentric all interventions must be standardized.

Author Response

Reviewer 1                       

Well designed paper. However the sample size and high dropout rates make results less reliable.

Thank you for this important comment. Reference to the small sample size was given in the prior manuscript on p.19 in the discussion section:

The sample size is too minor for small effects and should be supplemented by a control group for a clinically valid statement.”

A note regarding the classification and evaluation of the results in terms of sample size and drop outs was missing and is now given on p.21, in the discussion The revision is as follows:

It should also be noted that in Germany by the summer of 2022, some regulations regarding the COVID-19 pandemic had already been repealed, no school closures were occurring at that time, and leisure programs were once again being offered, making comparison with first-phase studies of the pandemic challenging. Due to the small sample and the high drop-out rate, the results might have a lower reliability and should be considered as preliminary findings.”

Yoga interventions are not clearly mentioned.

Thank you very much for this addition, this note brought important supplements to the intervention specification. On page 9 in methods section, the description of the yoga intervention has now been supplemented with more detailed information, as follows:

The perception and connection of the physical reactions of one's own limits will be brought closer by means of breathing exercises, balance exercises (e.g. Om mantra, focus on exhalation (Rechaka)), in the form of asanas (e.g. crane (Bakasana), chair (Utkatasana), mountain hold (Tatsana)) and concentration exercises, all embedded in an ice world imaginary journey.”

Being multicentric all interventions must be standardized.

Thank you for this significant comment. In this study, all interventions were standardized. The existing text references in the submitted manuscript can be found on page 9 (methods section):

The procedure of the activities followed a standardized manual, which was implemented in each institution.”

As well as on page 20:

“We can only speculate about reasons for this effect: as the face-to-face program was provided in a standardized manner, the children might have used the app more often, implying a higher intervention dose.

Reviewer 2 Report

I believe the study focus, objectives, design and analysis of data are appropriate to meet the criterion for scientific publication. As a preliminary study, the study provides useful evidence to promote adolescent mental health.

I would suggest adding some points in the discussion about following factors. 

Social and cultural factors that may have influenced because study findings indicated that being female, being in psycho-social crisis or migration were relevant. Need to further elaborate how these social factors relate with cultural factors and then overall increase the risk for poor mental health.

Some discussion points can be added to demonstrate how technology literacy and conveince with use of online mental app might be relevant. 

Add some directions for future research. 

Author Response

Reviewer 2:

Social and cultural factors that may have influenced because study findings indicated that being female, being in psycho-social crisis or migration were relevant. Need to further elaborate how these social factors relate with cultural factors and then overall increase the risk for poor mental health.

Thank you very much for this important input, this connection was not explicitly mentioned in the first version of the manuscript but is an important addition to put the findings into this context. Therefore, we added the following extension to the manuscript (see p.19 and p.20, discussion section):

“First, associations between prior identified risk factors and the baseline scores regarding psychosocial distress such as gender [8], having mentally ill parents [6], migration background [7] and a current crisis, were examined. Except for current crisis, the hypothesized correlates could not be confirmed by the data. Being female, having a migration background, and parental mental illness could not be replicated as risk factors in this study based on the baseline assessment.

On page 20, a further supplement referring to sociocultural factors was now added:

“However, it should be noted that these factors may be interrelated and are additionally influenced by socio-cultural factors. For instance, shame can be a socio-cultural factor which might influence reporting of psychological well-being, as shown in [54]. Thus, socio-cultural factors might have an additional influence on the study findings. As we did not assess further socio-cultural factors, the hypothesized relation could not be examined in this study.”

Some discussion points can be added to demonstrate how technology literacy and conveince with use of online mental app might be relevant.

Thank you very much for this significant impulse, which has now been included (see p. 19 in the discussion):

“Although the participants' media usage behavior was assessed, no information was gathered regarding their own smartphone possession or their experiences with and affinity for mental health apps. These previous experiences might add to the participants’ technologic and eHealth literacy. This, in turn, might beneficially impact the acceptance and usage of a mental health app [37, 38].

Add some directions for future research.

Thank you very much for this important addition, which was given too little scope in the previous manuscript. Additions to future directions of studies were now given on page 21 in the discussion section. We now changed the structure of the last part of the discussion to best include these added directions for future research. The revised section now reads as follows:

Future studies regarding blended intervention designs should include a larger sample, a randomized controlled design with a comparison group, and follow-up measurements in order to determine statistically valid conclusions regarding efficacy. This intervention included multiple therapeutical components and implementation formats. A next step would be to examine these components and delivery formats concerning their respective influence on the effectiveness using a dismantling study. Further research should also examine the role of socio-cultural aspects as well as technological and e-Mental health literacy on the feasibility and effectiveness of digital interventions for children and adolescents.”

Another additional reference was drawn on page 21 below:

Nevertheless, children and adolescents are currently experiencing an accumulation of global crises such as climate change, the Ukraine war, and currently the feminist revolution in Iran. A focus on youth’s coping in those global crises and the development and implementation of crisis-related interventions would be desirable in future research and clinical practice.”